# MvHSTM: A Multi-view Hypergraph Spatio-Temporal Model for Traffic Speed Forecasting

## Abstract

Accurate traffic speed prediction is critical in modern society as it is effective for both individuals and authorities. Due to the large scale of urban road networks, traffic speed exhibits complex spatio-temporal dependencies, not only among adjacent nodes but also across the network, reflecting both local and cross-regional simultaneous correlations. However, existing studies have not effectively addressed these characteristics. In this context, we propose a novel framework called Multi-view Hypergraph Spatio-Temporal Model (MvHSTM) that employs a temporal transformer to capture temporal dependencies and utilizes hypergraph convolutional networks to inherently model spatial relationships. Specifically, we introduce two hypergraph construction methods, the Geographical Adjacency Hypergraph (GAH) and the Feature Similarity Hypergraph (FSH), to capture spatial correlations on neighboring and non-neighboring scales. Extensive experiments on real-world traffic speed datasets demonstrate that our approach achieves state-of-the-art performance compared to baseline methods.

## 1 Introduction

Traffic speed prediction is a critical component in modern society, with applications in urban traffic management and intelligent transportation systems (ITS). Accurate forecasting of traffic speeds optimizes route navigation, improves estimated time of arrival, and enables efficient traffic management. Laor & Galily (2022) note that around 74 percent of smartphone users rely on navigation software, with the real-time "recalculating route" feature being particularly useful in offering reassurance and generating positive emotions. The significance of this predictive capability also extends to various domains. In navigation systems, accurate predictions facilitate optimal route selection, potentially reducing travel times and alleviating congestion. Emergency services benefit from more precise estimation of response times. Furthermore, improved traffic prediction contributes to environmental sustainability by promoting more efficient transportation patterns and potentially reducing carbon emissions.

However, the complexity of urban traffic systems necessitates sophisticated modeling approaches. Traffic is influenced by multiple factors, including spatial dependencies between road segments, temporal patterns such as rush hours. Correlations in the spatial dimension indicate congestion in one area can rapidly propagate to neighboring segments, creating ripple effects throughout the network. Besides, this kind of spatial similarity also far beyond simple geographical proximity. Road networks form complex webs of interconnected segments, where the speed can significantly drop after merging on freeways or dramatically rise back after diverging. Moreover, certain road segments may have functional relationships that are not immediately apparent from their geographical adjacency. For example, parallel arterial roads usually serve as alternatives, sharing traffic load during congestion. Expressways linking urban and suburban areas often exhibit synchronized congestion patterns during rush hours. Different areas within a city can have vastly different traffic characteristics due to land use patterns, population density, or the presence of key destinations. Therefore, developing comprehensive models that can capture these intricate spatial relationships along with their temporal dynamics is crucial for accurate traffic speed prediction. Such models need to learn and represent the different characteristics of road segments in the spatial dimension, presenting a significant challenge in the field of traffic forecasting.

Existing approaches to traffic prediction mainly focus on two aspects: time series methods and graph methods, corresponding to the temporal and spatial dimensions, respectively. Classical statistical methods such as Autoregressive Integrated Moving Average (ARIMA) (Williams & Hoel, 2003) and Support Vector Regression (SVR) (Drucker et al., 1996) primarily focus on modeling the sequential nature of traffic patterns over time. However, these approaches face limitations in capturing complex nonlinear relationships and spatial correlations inherent in traffic data. Recently, researchers have employed deep learning methods to address these shortcomings. Kim et al. (2017) introduce convolutional operations into Fully Connected LSTM (FC-LSTM) (Graves, 2013), which enables the original temporal model to perceive spatial relations. (Zhang et al., 2017) employ CNN in residual learning blocks, effectively addressing spatio-temporal correlations in urban grid pedestrian data. However, RNNs (including LSTM) have difficulty in capturing long-term temporal dependencies and are computationally intensive. Existing CNNs, while good at extracting grid-like spatial feature, struggle to model irregular road systems in real cities.

In order to model real road networks, graph-structured models are applied in recent studies. STGCN (Yu et al., 2017) applies graph convolutions and gated temporal convolutions to learn spatial and temporal dependencies, respectively. By applying a diffusion process on graph structure with RNN, DCRNN (Li et al., 2017) effectively model spatio-temporal network in traffic datasets. Extensive research, such as Graph WaveNet (Wu et al., 2019), STSGCN (Song et al., 2020), AGCRN (Bai et al., 2020) has proved that the effectiveness of graph structure in modeling real road networks. These studies have proved the effectiveness of graph structure in modeling urban road networks, relating segments by linking nearby nodes with edges. Nevertheless, standard graphs represent spatial relationships with simple pairwise edges, limiting their ability to capture complex interactions among multiple nodes. This structure makes it challenging to model higher-order spatial correlations and transregional relationships in traffic data. Furthermore, a large number of computations are needed to establish connections between distant nodes. In contrast, hypergraphs allow hyperedges to connect multiple nodes simultaneously, enabling a more comprehensive representation of spatial dependencies. Feng et al. (2019) extends traditional graph neural networks to hypergraphs, allowing the model to capture higher-order relationships. Wang et al. (2021) employ hypergraph on metro system, validating the performance in extracting higher-order relationships. Hypergraph The attention mechanism (Vaswani, 2017) also demonstrates its superiority in both spatial and temporal features, such as ASTGCN (Guo et al., 2019), STTN (Xu et al., 2020), and GMAN (Zheng et al., 2020). These prior works have significantly advanced the field of spatio-temporal traffic prediction by demonstrating the effectiveness of graph-based and attention-enhanced models in modeling road networks. The introduction of hypergraphs has further enriched spatial modeling by capturing higher-order relationships, which are difficult to model using standard graphs. However, existing hypergraph construction methods have certain limitations, as predefined rules may not fully capture the complex nature of traffic data.

To address these limitations, we propose a novel framework, the Multi-view Hypergraph Spatio-Temporal Model (MvHSTM), designed to capture the spatio-temporal features in traffic systems. Our method constructs hypergraph in two separate strategies, by geographical adjacency and traffic pattern similarity, respectively. The model utilizes a transformer module to handle temporal relationships, and two hypergraph convolution networks, Geographical Adjacency Hypergraph (GAH) and Feature Similarity Hypergraph (FSH), to represent both neighboring spatial relationships and non-neighboring feature similarities. This enables the capture of higher-order spatial correlations that are often overlooked by simple graph-based models. Finally, we design a self-adaptive fusion module to obtain the prediction result.

Our contributions can be summarized as follows:

- We propose a novel MvHSTM framework to comprehensively capture the spatio-temporal features of traffic speed forecasting. The framework employs a temporal embedding for temporal transformer to model temporal features, and two hypergraph convolution networks are utilized to capture inherent spatial relationships.

- To represent spatial feature specifically in traffic speed data, we propose two different hypergraph construction approaches, forming the Geographical Adjacency Hypergraph (GAH) and the Feature Similarity Hypergraph (FSH). The GAH is constructed from nodes with their adjacencies, and the FSH is constructed from nodes with similar traffic speed patterns.

- We evaluate our MvHSTM in two different real-world traffic speed datasets and the results demonstrate that MvHSTM performs better than baselines.

## 2 PRELIMINARIES

The network of a road system can be defined as a graph $\mathcal{G} = (\mathcal{V}, \mathcal{E}, \boldsymbol{A})$, where $\mathcal{V}$ is a set of vertices representing speed sensors on roads, $\mathcal{E}$ is a set of edges representing roads linking sensors, and $\boldsymbol{A} \in \mathbb{R}^{N \times N}$ is an adjacency matrix that demonstrates relationships between vertices. On the basis of graph structure, a hypergraph consists of a set of vertices, and a set of hyperedges that link more than two vertices. We can define a hypergraph as $\mathcal{G}_h = (\mathcal{V}_h, \mathcal{E}_h, \boldsymbol{H})$, where $\mathcal{V}_h$ is the set of $N$ vertices, $\mathcal{E}_h$ is the set of $M$ hyperedges, and $\boldsymbol{H} \in \mathbb{R}^{N \times M}$ is the incidence matrix. The incidence matrix of hypergraph is defined as follows:

$$\boldsymbol{H}_{ij} = \begin{cases} 1 & \text{if } v_i \in e_j, \\ 0 & \text{if } v_i \notin e_j. \end{cases}$$

The time series of traffic speed is represented as $\mathbf{X}_{1:\tau} = (\boldsymbol{X}_1, \boldsymbol{X}_2, \ldots, \boldsymbol{X}_\tau) \in \mathbb{R}^{N \times T \times F}$, where $N$ is the number of vertices, $T$ is the sequence length, and $F$ is the dimension of features.

Given traffic time series $\mathbf{X}_{\tau-T+1:\tau} = (\boldsymbol{X}_{\tau-T+1}, \boldsymbol{X}_{\tau-T+2}, \ldots, \boldsymbol{X}_\tau)^T \in \mathbb{R}^{N \times T \times F}$ and road system $\mathcal{G} = (\mathcal{V}, \mathcal{E}, A)$, the problem of traffic forecasting can be formulated as follows:

$$\hat{\mathbf{X}}_{\tau+1:\tau+T} = f(\mathbf{X}_{\tau-T+1:\tau}, \mathcal{G})$$

where $T$ is the sequence length of the input series and the predict length.

## 3 METHODOLOGY

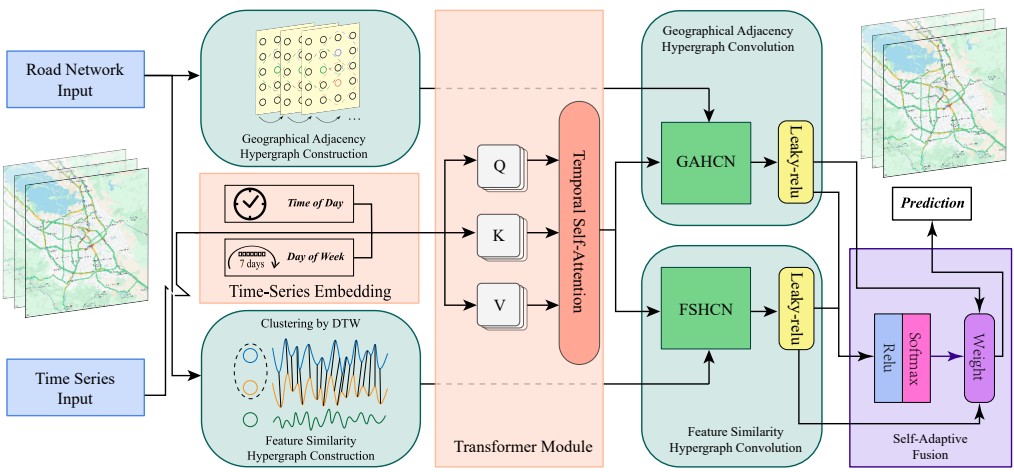

Figure 1: The framework of the proposed Multi-view Hypergraph Spatio-Temporal Model (MvHSTM). It consists of time-series embedding, a transformer module to handle temporal dependencies, and the construction and convolution of a Geographical Adjacency Hypergraph as well as a Feature Similarity Hypergraph.

In this section, the components of MvHSTM are represented detailedly. As shown in Figure 1, the input data consists of two parts, the road network input is processed to construct two hypergraphs that demonstrate spatial correlations, and the time series input goes through temporal embedding to represent temporal features. The embedded data is further processed by the transformer module,

to extract temporal features from each time series. After that, time series data is associated with the two constructed hypergraphs, and is further processed by the Geographical Adjacency Hypergraph Convolution Network (GAHCN) and the Feature Similarity Hypergraph Convolution Network (FSHCN), respectively. As the two hypergraphs represent different inherent correlations, we fuse the two results using a self-adaptive weight, and finally obtain the prediction.

## 3.1 TRANSFORMER MODULE

**Time-series Embedding** To recognize temporal feature of traffic speed data, we utilize a temporal embedding layer to extract time-of-day feature and day-of-week feature within data. The time-of-day feature $\boldsymbol{T}_{tod}$ ranging from 1 to 288, representing 288 timestamps, and the day-of-week feature $\boldsymbol{T}_{dow}$ ranging from 1 to 7, corresponding to Monday to Sunday. These temporal information is processed by the temporal embedding layer to obtain the feature embedding $\mathbf{E}_f \in \mathbb{R}^{N \times T \times (F+2 \times d_f)}$:

$$\mathbf{E}_f = [\mathbf{X}; FC_{tod}(\mathbf{T}_{tod}); FC_{dow}(\mathbf{T}_{dow})]$$

where $d_f$ is the dimension of each feature embedding, and $FC$ are fully connected layers.

**Temporal Self-attention** The temporal self-attention mechanism (Jiang et al., 2023) is designed to capture the temporal dependencies within the traffic time series data. Given the tensor $\mathbf{X} \in \mathbb{R}^{N \times T \times d_h}$, where $N$ is the number of vertices, $T$ is the number of time frames, and $d_h$ is the hidden dimension, the temporal self-attention layer computes the query, key, and value matrices as follows:

$$\boldsymbol{Q}_t = \boldsymbol{X}_{n,:,:}\boldsymbol{W}^Q, \boldsymbol{K}_t = \boldsymbol{X}_{n,:,:}\boldsymbol{W}^K, \boldsymbol{V}_t = \boldsymbol{X}_{n,:,:}\boldsymbol{W}^V,$$

where $\boldsymbol{W}^Q, \boldsymbol{W}^K, \boldsymbol{W}^V \in \mathbb{R}^{d_h \times d_h}$ are learnable parameters. Then, the self-attention scores are calculated as the following equation:

$$\boldsymbol{A}_{(t)} = Softmax(\frac{\boldsymbol{Q}_t \boldsymbol{K}_t^\top}{\sqrt{d_h}}),$$

Finally, the output of transformer module is calculated as follows:

$$\boldsymbol{Z}_{(\mathrm{t})} = \boldsymbol{A}_{(\mathrm{t})}\boldsymbol{V}_{(\mathrm{t})}.$$

## 3.2 CONSTRUCTION OF HYPERGRAPH

For a graph $\mathcal{G} = (\mathcal{V}, \mathcal{E}, \boldsymbol{A})$, a critical step in constructing a hypergraph $\mathcal{G}_h = (\mathcal{V}_h, \mathcal{E}_h, \boldsymbol{H})$ is to decide which vertices $\in \mathcal{V}$ should be linked by each hyperedge $\in \mathcal{E}_h$. In this subsection, we introduce two separate approaches for assembling vertices into hyperedges, to obtain a Geographical Adjacency Hypergraph (GAH) and a Feature Similarity Hypergraph (FSH).

**Construction of GAH** Normally, a hypergraph is applied in traffic forecasting for its effectiveness in gathering adjacency road segments in a complex enormous road system. The Geographical Adjacency Hypergraph (GAH) uses this characteristic. In constructing the GAH, we utilize the input adjacency matrix $\boldsymbol{A} \in \mathbb{R}^{N \times N}$, where each entry $A_{ij}$ represents the spatial connectivity between vertices $i$ and $j$. Each hyperedge is constructed from a vertex $v \in \mathcal{V}$ and its $k$-nearest neighboring vertices based on their adjacency in the road system:

$$\mathcal{E}_h^{GAH} = \{e_h^v : e_h^v = \{v\} \cup \mathrm{N}_k(v)\}$$

where $\mathrm{N}_k(v)$ is the set of $k$ vertices that are most strongly connected to $v$ according to the adjacency matrix $\boldsymbol{A}$.

**Constructing Hyperedges by DTW** Dynamic Time Warping (DTW) (Berndt & Clifford, 1994) is an algorithm designed to measure the similarity between two temporal sequences. Unlike simple Euclidean distance, DTW aligns time series flexibly by permitting shifts in the time dimension, thereby offering a more accurate similarity measure even when the sequences are not perfectly synchronized.

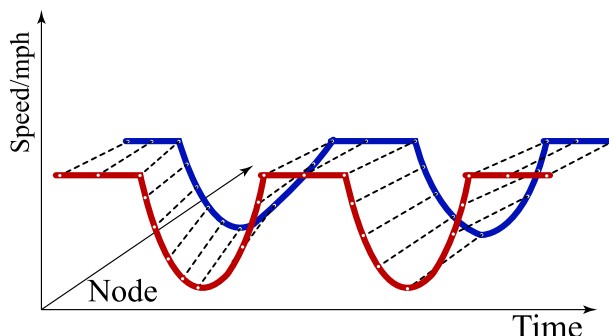

Figure 2: The DTW algorithm measures the similarity between two time series by aligning them. This algorithm can effectively cluster time series with similar patterns.

Given two sequences $\boldsymbol{x} = (x_1, x_2, \ldots, x_N)$ and $\boldsymbol{y} = (y_1, y_2, \ldots, y_M)$, the objective is to find the warping path $\boldsymbol{w} = (w_1, w_2, \ldots, w_L)$, where each element $w_l = (i, j)$ represents an optimal alignment between the $i$-th point of $\boldsymbol{x}$ and the $j$-th point of $\boldsymbol{y}$. This path minimizes the cumulative alignment cost, which can be expressed as follows.

$$\text{DTW}(\boldsymbol{x}, \boldsymbol{y}) = \min_W \sum_{l=1}^{L} d(w_l)$$

where $d(w_l)$ is the Euclidean distance between aligned points in the sequences.

Hypergraphs are an extension of traditional graphs, where an edge (namely a hyperedge) can connect multiple vertices, allowing for a more expressive representation of detailed variation patterns among the vertices. By applying DTW, stations with similar temporal patterns are grouped into the same hyperedge. In a real traffic speed datasets, factors like rush hour and daily routines cause non-linear variations, and two cyclical patterns are particularly significant: the daily cycle and the weekly cycle.

Taking a week period from Jan 9, 2017 to January 16, 2017 in the PEMS-BAY dataset as an example, we utilize the DTW algorithm to cluster 325 vertices into 16 different patterns. As shown in Figure 3, three typical cases of hyperedges represent three different traffic speed patterns, and vertices with the same patterns are distributed to throughout the city. The traffic speeds on sensors in Hyperedge A exhibits a clear drop in morning and evening peak hours on weekdays (Monday to Friday). These sensor stations are heavily influenced by daily commuting patterns and speeds drop to about 35 mph (half of free flow speed, reaching capacity of freeway segments). In contrast, the traffic speeds of sensors in Hyperedge B are relatively stable throughout the week, with minimal fluctuations. Hyperedge B is a typical case of vertices with consistent traffic flow, potentially unaffected by typical rush hour variations. Traffic speeds in Hyperedge C dip once per day to about 15 mph in weekdays, indicating they have only one peak hour but effect capacity seriously. This indicates that same pattern can happen in different region of the city. By employing DTW, these correlations can be discerned, contributing to more precise traffic speed prediction.

**Construction of FSH** The FSH is constructed by clustering vertices based on their traffic speed patterns. Traffic speed data exhibits different patterns across various scenarios, such as commuting, tourism, and holidays, while also demonstrating periodicity over different cycles (e.g., daily, weekly). We use a seven-days period covering the cycles to recognize traffic variation patterns among the vertices. To achieve this, we apply the DTW algorithm, which measures the similarity between two temporal sequences by aligning them in a way that minimizes the difference.

$$\mathcal{E}_h^{FSH} = \{e_h^c : e_h^c = \text{Cluster}_c(V), \ c = 1, \ldots, k\}$$

where $\text{Cluster}_c(V)$ is the set of vertices grouped into cluster $c$ based on their speed sequence similarity using the DTW algorithm.

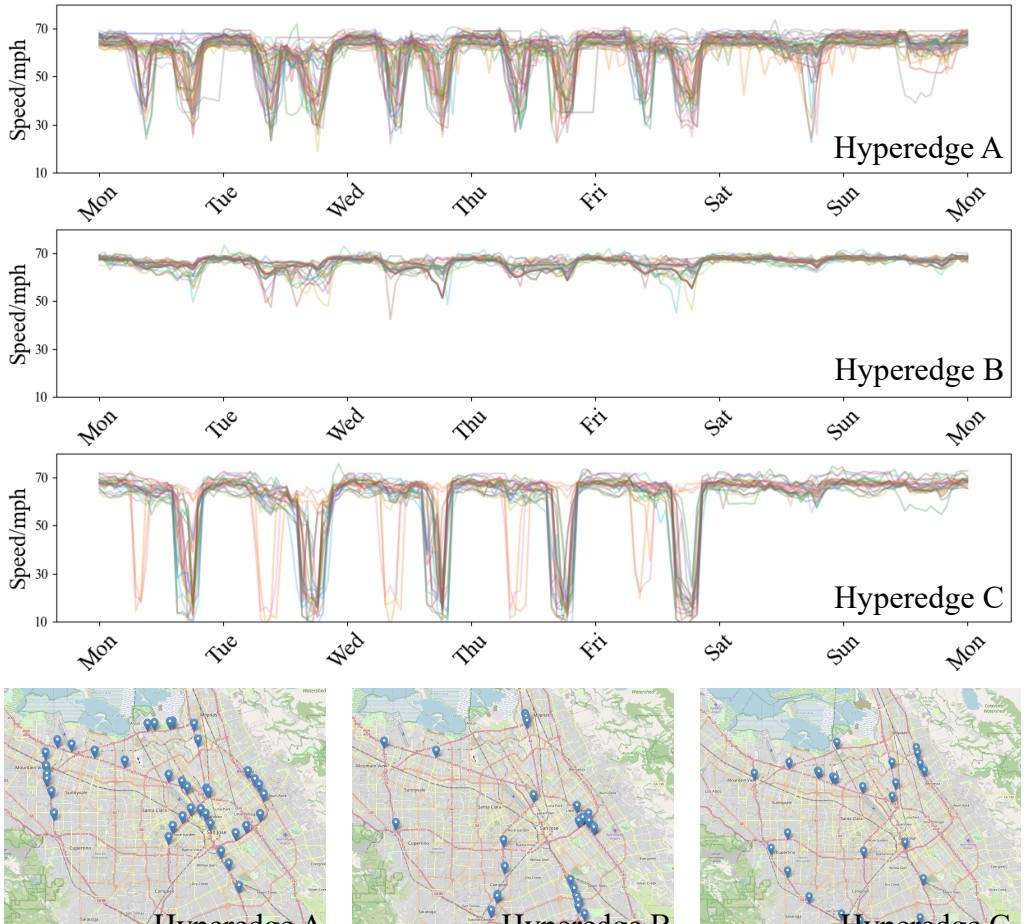

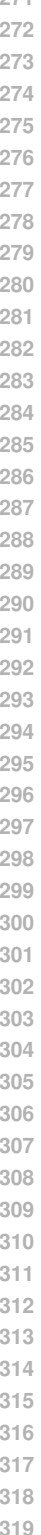

Figure 3: Temporal speed patterns and spatial distribution for hypergraph construction. The top three panels illustrate the traffic speed variations for three hyperedges (A, B, and C) over the week. Hyperedge A exhibits high-speed fluctuations throughout weekdays. Hyperedge B displays more stable speed trends throughout the entire week. Hyperedge C shows distinct speed drops during peak hours, especially on weekdays. The bottom panels show the spatial distribution of the nodes forming each hyperedge.

### 3.3 HYPERGRAPH CONVOLUTION

**Hypergraph Convolution** After constructing the hypergraph, we proceed to define the hypergraph convolution operation. The convolution process is utilized on both GAH and FSH, namely Geographical Adjacency Hypergraph Convolutional Network (GAHCN) and Feature Similarity Hypergraph Convolutional Network (FSHCN). This operation is crucial for processing and propagating information across the hyperedges and vertices. The hypergraph convolution can be expressed as follows:

$$\mathbf{X}_i^{(l+1)} = D_v^{-\frac{1}{2}} HW D_e^{-1} H^\top D_v^{-\frac{1}{2}} \mathbf{X}_i^l \Theta$$

where $\mathbf{D}_v$ is the vertex degree matrix, $\mathbf{D}_e$ is the hyperedge degree matrix, $\mathbf{W}$ is the importance of each hyperedge, and $\Theta$ is the parameter to be learned in the model.

**Self-adaptive Fusion** To effectively integrate features extracted from two different hypergraphs, a self-adaptive fusion strategy is employed. This mechanism adaptively learns the importance of each

feature set based on their contributions to the overall task, and computing the weight of each feature. After that, the two results $\mathbf{X}_1$ and $\mathbf{X}_2$ are fused using the calculated weights to predict the future series of traffic speeds:

$$[\mathbf{W}_1; \mathbf{W}_2] = Softmax\left(FC\left(ReLU\left(FC([\mathbf{X}_1; \mathbf{X}_2])\right)\right)\right)$$

$$\mathbf{X}_{fused} = \prod_{i=1}^{2}(\mathbf{X}_i\mathbf{W}_i)$$

## 4 EXPERIMENTS

### 4.1 EXPERIMENTAL SETUP

In this section, we employed two widely-used traffic datasets, METR-LA and PEMS-BAY (Li et al., 2017), to validate our proposed model. Both datasets divide the traffic speed data into 5-minute intervals. METR-LA contains traffic data from March to June 2012, and PEMS-BAY ranges from January to May 2017. The two datasets are divided into training, validation, and test sets with a ratio of 7:1:2, respectively. The descriptions of datasets are shown in Table 1:

Table 1: Dataset description.

| Dataset | Sensors | Time Steps | Missingness |
|---------|---------|-----------|-------------|
| METR-LA | 207 | 34272 | 8.11% |
| PEMS-BAY | 325 | 52116 | 0.003% |

The input and prediction lengths are set to 12 time steps, corresponding to one hour. For temporal transformer, the feature embedding dimension of time-of-day and day-of-week are set to 24, and the adjusted embedding dimension is configured to 80. The number of layers for temporal transformer is set to 3, and each equipped with 4 attention heads. In terms of hypergraph construction, the number of nearest k vertices in GAH is set to 4, and the number of hyperedges in FSH is set to 16. The number of layers for GAHCN and FSHCN are both set to 3. We utilize Adam as the optimizer, initialize a learning rate of 0.001 with learning rate decaying during the training. The batch size is set to 16, and early stop is employed to halt the training if the validation loss shows no improvement for 30 consecutive epochs. Early stopping is employed to halt the training if the validation loss showed no improvement for 30 consecutive epochs, and the maximum epoch is set to 100. The performance is evaluated on horizon 3, 6, and 12 by three metrics in time-series forecasting tasks: Mean Absolute Error (MAE), Root Mean Square Error (RMSE), and Mean Absolute Percentage Error (MAPE).

In this experiment, we compared our proposed MvHSTM model against multiple baselines, ranging from classic models to recent state-of-the-art approaches: HI (Historical Inertia), ARIMA (Auto Regressive Integrated Moving Average) (Makridakis & Hibon, 1997), FC-GAGA (Fully Connected Gated Graph Architecture) (Oreshkin et al., 2021), Graph Wavenet (Wu et al., 2019), DCRNN (Diffusion Convolutional Recurrent Neural Network) (Li et al., 2017), AGCRN (Adaptive Graph Convolutional Recurrent Network) (Bai et al., 2020), STGCN (Spatio-temporal Graph Convolutional Network) (Yu et al., 2017), STSGCN (Spatial-Temporal Synchronous Graph Convolutional Networks) (Song et al., 2020), GTS (Shang et al., 2021), MTGNN (Wu et al., 2020), STNorm (Spatial and Temporal Normalization) (Deng et al., 2021), GMAN (Graph Multi-Attention Network) (Zheng et al., 2020), STID (Spatial and Temporal IDentity) (Shao et al., 2022), PDFormer (Jiang et al., 2023).

### 4.2 EXPERIMENT RESULTS

Table 2 presents the performance comparison of various traffic prediction models on two benchmark datasets. MvHSTM outperforms baseline models on both METR-LA and PEMS-BAY datasets. Other transformer-based models such as PDFormer, GMAN and graph-based model such as DCRNN also perform competitively. These results demonstrate the efficacy of our model's multi-view hypergraph structure, which adeptly models both neighboring and non-neighboring traffic patterns, addressing the limitations of former graph-based models. This capability enables MvHSTM to outperform its counterparts by capturing higher-order relationships and intricate interactions within the traffic network, which are often elusive to conventional graph-based models.

Table 2: Performance on METR-LA and PEMS-BAY

| Datasets | Models | 15 min | | | 30 min | | | 60 min | | |
|---|---|---|---|---|---|---|---|---|---|---|
| | | MAE | RMSE | MAPE(%) | MAE | RMSE | MAPE(%) | MAE | RMSE | MAPE(%) |
| METR-LA | HI | 6.80 | 14.21 | 16.72 | 6.80 | 14.21 | 16.72 | 6.80 | 14.20 | 10.15 |
| | ARIMA | 3.99 | 8.21 | 9.60 | 5.11 | 10.45 | 12.70 | 6.90 | 13.23 | 17.40 |
| | FC-GAGA | 2.75 | 5.34 | 7.25 | 3.10 | 6.30 | 8.57 | 3.51 | 7.31 | 10.14 |
| | GWNet | 2.69 | 5.15 | 6.99 | 3.08 | 6.20 | 8.47 | 3.51 | 7.28 | 9.96 |
| | DCRNN | 2.67 | 5.16 | 6.86 | 3.12 | 6.27 | 8.42 | 3.54 | 7.47 | 10.32 |
| | AGCRN | 2.85 | 5.53 | 7.63 | 3.20 | 6.52 | 9.00 | 3.59 | 7.45 | 10.47 |
| | STGCN | 2.75 | 5.29 | 7.10 | 3.15 | 6.35 | 8.62 | 3.60 | 7.43 | 10.35 |
| | STSGCN | 3.31 | 7.62 | 8.06 | 4.13 | 9.77 | 10.29 | 5.06 | 11.66 | 12.91 |
| | GTS | 2.75 | 5.27 | 7.12 | 3.14 | 6.33 | 8.62 | 3.59 | 7.44 | 10.25 |
| | MTGNN | 2.69 | 5.16 | 6.89 | 3.05 | 6.13 | 8.16 | 3.47 | 7.21 | **9.70** |
| | STNorm | 2.81 | 5.57 | 7.40 | 3.18 | 6.59 | 8.47 | 3.57 | 7.51 | 10.24 |
| | GMAN | 2.80 | 5.55 | 7.41 | 3.12 | 6.49 | 8.73 | 3.44 | 7.35 | 10.07 |
| | STID | 2.82 | 5.53 | 7.75 | 3.19 | 6.57 | 9.39 | 3.55 | 7.55 | 10.95 |
| | PDFormer | 2.83 | 5.45 | 7.77 | 3.20 | 6.46 | 9.19 | 3.62 | 7.47 | 10.91 |
| | MvHSTM | **2.62** | **5.03** | **6.72** | **2.96** | **6.00** | **8.11** | **3.40** | **7.15** | 9.95 |
| PEMS-BAY | HI | 3.06 | 7.05 | 6.85 | 3.06 | 7.04 | 6.84 | 3.05 | 7.03 | 6.83 |
| | ARIMA | 1.62 | 3.30 | 3.50 | 2.30 | 4.76 | 5.40 | 3.38 | 6.50 | 8.30 |
| | FC-GAGA | 1.36 | 2.86 | 2.87 | 1.80 | 3.80 | 3.80 | 3.51 | 7.31 | 10.14 |
| | GWNet | 1.30 | 2.73 | 2.71 | 1.63 | 3.73 | 3.73 | 1.99 | 4.60 | 4.71 |
| | DCRNN | 1.31 | 2.76 | 2.73 | 1.65 | 3.75 | 3.71 | 1.97 | 4.60 | 4.68 |
| | AGCRN | 1.35 | 2.88 | 2.91 | 1.67 | 3.82 | 3.81 | 1.94 | 4.50 | 4.55 |
| | STGCN | 1.36 | 2.88 | 2.86 | 1.70 | 3.84 | 3.79 | 2.02 | 4.63 | 4.72 |
| | STSGCN | 1.44 | 3.01 | 3.01 | 1.83 | 4.18 | 4.17 | 2.26 | 5.21 | 5.40 |
| | GTS | 1.37 | 2.92 | 2.85 | 1.72 | 3.86 | 3.88 | 2.06 | 4.60 | 4.88 |
| | MTGNN | 1.33 | 2.80 | 2.81 | 1.66 | 3.77 | 3.75 | 1.95 | 4.50 | 4.62 |
| | STNorm | 1.33 | 2.82 | 2.76 | 1.65 | 3.77 | 3.66 | 1.92 | 4.45 | **4.46** |
| | GMAN | 1.35 | 2.90 | 2.87 | 1.65 | 3.82 | 3.74 | 1.92 | 4.49 | 4.52 |
| | STID | 1.31 | 2.79 | 2.78 | 1.64 | 3.73 | 3.73 | 1.91 | 4.42 | 4.55 |
| | PDFormer | 1.32 | 2.83 | 2.78 | 1.64 | 3.79 | 3.71 | 1.91 | 4.43 | 4.51 |
| | MvHSTM | **1.30** | **2.73** | **2.74** | **1.61** | **3.64** | **3.64** | **1.90** | **4.35** | 4.49 |

## 4.3 ABLATION STUDY

To further evaluate the effectiveness of each component proposed in MvHSTM, we conduct experiments using four variations of original MvHSTM on the METR-LA dataset:

- **w/o $E_t$**: This configuration removes the temporal embedding $E_t$.
- **w/o GAH**: This excludes the Geographical Adjacency Hypergraph component.
- **w/o FSH**: This excludes the Feature Similarity Hypergraph component.
- **w/o FSH & GAH**: Both GAH and FSH are excluded in this configuration.
- **MvHSTM**: The complete version of the MvHSTM.

Table 3 demonstrates the results of each variation. $E_t$ is designed to represent time-of-day and day-of-week within traffic data, which mainly contribute to temporal dimension of traffic prediction. Specifically, after removal $E_t$, the mean of three errors in 15 minutes, 30 minutes, and 60 minutes increases by 3.7%, 4.6% and 3.7%, respectively. GAH and FSH are designed to capture spatial correlations at a neighboring scale and similar but non-neighboring scale, respectively. The results indicate that model without either FSH or GAH performs the worst. GAH component contributes to spatial contributes significantly to capturing spatial correlation, and FSH can further increase the accuracy. Specifically, the worst performance can be seen in the model without FSH & GAH, with errors increasing by 9.9%, 9.5%, 3.6% on three horizons. Without GAH, the average increase in the three errors across the 15, 30, and 60-minute horizons is 8.5%, 7.7%, and 4.6%, respectively, indi-

cating that GAH plays a significant role in improving model performance by capturing the spatial dependencies effectively. While FSH has a noticeable effect on reducing errors, its impact is less pronounced than GAH. However, it still enhances the model's capability to learn feature similarities among nodes, particularly for shorter horizons. This indicates that adjacency correlations are important for the spatial dimension in traffic prediction, and the combination of GAH and FSH benefit the model to capture spatial correlation the most. Therefore, all the components are necessary for temporal or spatial features.

Table 3: Ablation Study on METR-LA.

| Configuration | 15min | | | 30min | | | 60min | | |
|---|---|---|---|---|---|---|---|---|---|
| | MAE | RMSE | MAPE(%) | MAE | RMSE | MAPE(%) | MAE | RMSE | MAPE(%) |
| w/o $E_t$ | 2.72 | 5.17 | 7.03 | 3.09 | 6.20 | 8.61 | 3.52 | 7.24 | 10.60 |
| w/o GAH | 2.78 | 5.49 | 7.40 | 3.13 | 6.47 | 8.89 | 3.53 | 7.44 | 10.53 |
| w/o FSH | 2.66 | 5.08 | 6.87 | 2.99 | 6.02 | 8.15 | 3.42 | 7.20 | 10.05 |
| w/o FSH & GAH | 2.82 | 5.57 | 7.48 | 3.06 | 6.78 | 9.10 | 3.34 | 7.57 | 10.63 |
| MvHSTM | **2.62** | **5.03** | **6.72** | **2.96** | **6.00** | **8.11** | **3.40** | **7.15** | **9.95** |

## 4.4 PARAMETER SENSITIVITY ANALYSIS

A crucial parameter in MvHSTM is the number of hyperedges used for constructing the FSH. The choice of this parameter directly impacts how many different patterns are discerned in capturing spatial correlations in the traffic network. To analyze the sensitivity of MvHSTM to the number of hyperedges, we vary this parameter and evaluate the model's performance on the METR-LA dataset, focusing on metrics as the number of hyperedges ranges from 8 to 20.

Table 4: Parameter Sensitivity Analysis on METR-LA.

| Number of Hyperedge | 15min | | | 30min | | | 60min | | |
|---|---|---|---|---|---|---|---|---|---|
| | MAE | RMSE | MAPE(%) | MAE | RMSE | MAPE(%) | MAE | RMSE | MAPE(%) |
| 8 | 2.67 | 5.08 | 6.86 | 3.01 | 6.06 | 8.24 | 3.45 | 7.17 | 10.06 |
| 12 | 2.64 | 5.05 | 6.80 | 2.98 | 6.01 | 8.13 | 3.42 | 7.16 | 10.01 |
| 16 | **2.62** | **5.03** | **6.72** | **2.96** | **6.00** | **8.11** | **3.40** | **7.15** | **9.95** |
| 20 | 2.66 | 5.20 | 6.90 | 3.07 | 6.10 | 8.23 | 3.48 | 7.27 | 10.10 |

As shown in Table 4, setting the number of hyperedges to 16 is optimal for achieving high predictive accuracy in MvHSTM, providing a balance between capturing detailed spatial correlations and maintaining model efficiency across all prediction intervals. Fewer hyperedges may not capture spatial correlations precisely, but a greater number of hyperedges can also lead to decreased performance. Therefore, in this study, we use 16 hyperedges to model the traffic network.

## 5 CONCLUSION

In this study, we introduce the Multi-view Hypergraph Spatio-Temporal Model (MvHSTM) for traffic speed forecasting. By integrating a temporal transformer and two hypergraphs, the Geographical Adjacency Hypergraph (GAH) and the Feature Similarity Hypergraph (FSH), our model effectively captures complex spatio-temporal dependencies. Experiments indicate that applying the hypergraph construction method based on feature similarity is effective in traffic predicting. Tests on the METR-LA and the PEMS-BAY datasets show that MvHSTM outperforms state-of-the-art models, demonstrating its potential for improving traffic management and route planning in intelligent transportation systems.

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
