# OpenReview forum: "MvHSTM: A Multi-view Hypergraph Spatio-Temporal Model for Traffic Speed Forecasting"
_ICLR.cc/2025/Conference — ICLR 2025 Conference Withdrawn Submission_

### Official Review · Reviewer_piXm · 2024-10-30

**Soundness:** 3
**Presentation:** 3
**Contribution:** 1
**Rating:** 5
**Confidence:** 4

**Summary:**

The paper introduces the Multi-view Hypergraph Spatio-Temporal Model (MvHSTM) for traffic speed forecasting, presenting a unified framework that integrates temporal transformers with hypergraph convolutional networks to effectively capture both sequential and spatial dependencies in traffic data. By constructing two distinct hypergraphs - Geographical Adjacency Hypergraph (GAH) and Feature Similarity Hypergraph (FSH) - the model adeptly represents neighboring and feature-based non-neighboring spatial relationships. Comprehensive experiments on METR-LA and PEMS-BAY datasets demonstrate that MvHSTM achieves superior performance compared to a range of baseline models, highlighting its capability to model complex spatio-temporal interactions in urban traffic networks.

**Strengths:**

1. **Clarity and Detailed Introduction of the Methodology:**
The paper provides a thorough and well-structured explanation of the proposed MvHSTM framework. Each component, including the temporal transformer, Geographical Adjacency Hypergraph (GAH), Feature Similarity Hypergraph (FSH), and the self-adaptive fusion module, is meticulously described with clear mathematical formulations and illustrative figures. This comprehensive detailing facilitates a deep understanding of the model's architecture and the underlying mechanisms that enable effective spatio-temporal traffic forecasting.

2. **Unified Framework for Sequential and Spatial Pattern Modeling:**
MvHSTM successfully integrates temporal and spatial modeling within a single cohesive framework. The use of a temporal transformer allows the model to capture intricate temporal dependencies, including daily and weekly traffic patterns. Simultaneously, the incorporation of hypergraph convolutional networks enables the modeling of both local and global spatial relationships among traffic sensors. This joint modeling approach ensures that the model can leverage the full spectrum of spatio-temporal information inherent in traffic data, leading to more accurate and reliable speed predictions.

3. **Explainable Insights Through Constructed Hyperedges:**
The construction of hyperedges based on geographical adjacency and feature similarity not only enhances the model's predictive performance but also provides valuable interpretability. The Geographical Adjacency Hypergraph (GAH) captures spatial correlations among neighboring road segments, while the Feature Similarity Hypergraph (FSH) groups sensors with similar traffic patterns, regardless of their physical proximity. This dual hypergraph approach offers explainable insights into the underlying traffic dynamics, such as identifying regions with synchronized congestion patterns or alternative routes sharing traffic loads during peak hours.

**Weaknesses:**

1. **Limited Novelty in Methodological Approach:**
While the integration of hypergraph networks and transformer architectures for spatio-temporal modeling is well-executed, these concepts have been extensively explored in existing literature. The paper would benefit from a more explicit delineation of the novel contributions and distinct advantages of MvHSTM over prior models. Highlighting unique aspects, such as the specific hypergraph construction techniques or the self-adaptive fusion mechanism, in greater depth would better emphasize the paper’s originality and contribution to the field.
2. **Insufficient Variety of Experimental Datasets:**
The evaluation of MvHSTM is conducted on two widely-used datasets, METR-LA and PEMS-BAY. While these datasets are standard benchmarks in traffic forecasting research, incorporating additional datasets from different geographical regions or varying traffic conditions could provide a more comprehensive assessment of the model's generalizability and robustness. Expanding the experimental validation to include diverse datasets would strengthen the evidence supporting MvHSTM’s effectiveness across different urban traffic environments.
3. **Marginal Improvements and Inconsistent Performance Gains:**
Although MvHSTM outperforms several baseline models, the extent of performance improvement is relatively modest in some cases. Additionally, there are instances where MvHSTM does not consistently outperform all baseline methods, particularly in longer prediction horizons. Providing a more detailed analysis of scenarios where the model underperforms, along with potential reasons and strategies for mitigating these limitations, would offer a more balanced evaluation of MvHSTM’s strengths and areas for improvement.

**Questions:**

1. **Scalability and Computational Efficiency:**
How does MvHSTM scale with larger and more complex traffic networks in terms of computational resources and training time? Are there any strategies implemented or proposed to enhance the model’s scalability for real-time traffic forecasting in expansive urban areas?
2. **More Details about Hypergraph Construction:**
As the constructed hyperedges offer good interpretability, can the authors provide more detailed examples and insights about the hypergraph construction process, as well as how the constructed hypergraph could benefit practical traffic management applications?

---

### Official Review · Reviewer_eeyR · 2024-11-03

**Soundness:** 2
**Presentation:** 2
**Contribution:** 2
**Rating:** 3
**Confidence:** 4

**Summary:**

This paper proposes a Multi-view Hypergraph Spatio-Temporal Model (MvHSTM) for traffic speed forecasting. Specifically, MvHSTM first utilizes the temporal Transformer to capture temporal dependencies. Then geographical and semantic hypergraph-based convolution networks are utilized to extract spatial correlations. Experiments on two traffic speed datasets demonstrate the state-of-the-art performance of MvHSTM.

**Strengths:**

1.	This paper constructs the hypergraph from the geographical and semantic aspects.
2.	The proposed model achieves superior performance on two datasets.

**Weaknesses:**

1. Lack of novelty. The hypergraph-based spatio-temporal forecasting methods have been researched, including the semantic-based HGCN [1], geographical-based ST-HSL [2], and adaptive-based AIMST [3]. The MvHSTM proposed in this paper seems like a combination of previous models.

2. Many important traffic speed forecasting baselines are ignored in experiments: GNN-based D2STGNN[4], Transformer-based STAEformer [5], and Attention-based STWave [6].

3. The variant "w/o FSH & GAH" achieves better long forecasting performance in the ablation study, which is against in designing the hypergraph in the method.

[1] Hierarchical Graph Convolution Network for Traffic Forecasting

[2] Spatial-Temporal Hypergraph Self-Supervised Learning for Crime Prediction

[3] Adaptive and Interactive Multi-Level Spatio-Temporal Network for Traffic Forecasting

[4] Decoupled Dynamic Spatial-Temporal Graph Neural Network for Traffic Forecasting

[5] STAEformer: Spatio-Temporal Adaptive Embedding Makes Vanilla Transformer SOTA for Traffic Forecasting

[6] When spatio-temporal meet wavelets: disentangled traffic forecasting via efficient spectral graph attention networks

**Questions:**

Please see in weaknesses.

---

### Official Review · Reviewer_YGbW · 2024-11-05

**Soundness:** 2
**Presentation:** 3
**Contribution:** 3
**Rating:** 5
**Confidence:** 3

**Summary:**

This paper introduces a novel framework designed to enhance the accuracy of traffic speed prediction, which is vital for urban traffic management and intelligent transportation systems. The authors highlight the complexities of urban traffic networks, characterized by intricate spatio-temporal dependencies that existing models have struggled to capture effectively.

**Strengths:**

The paper presents a novel framework, the Multi-view Hypergraph Spatio-Temporal Model (MvHSTM), which integrates hypergraph convolutional networks with a temporal transformer for traffic speed forecasting. This approach is original in its dual focus on both geographical adjacency and feature similarity, allowing for a more nuanced representation of complex spatio-temporal relationships in traffic data.

Hypergraphs allow for a more flexible representation of relationships among traffic sensors, capturing complex interactions that traditional graphs may overlook. The incorporation of a temporal transformer to model temporal dependencies is a significant advancement. By effectively capturing time-of-day and day-of-week features, the model acknowledges the inherent periodicity in traffic patterns.

The inclusion of ablation studies helps to isolate the impact of different components of the MvHSTM model. By demonstrating how each part contributes to the overall accuracy, the authors effectively highlight the importance of their approaches, such as the dual hypergraph construction methods.

**Weaknesses:**

Elaborating on the self-adaptive fusion mechanism used to combine outputs from the GAH and FSH would be important. Providing a detailed description of how weights are learned and applied during the fusion process would clarify how the model balances different sources of information.

I recommend including further discussion of technical details in the proposed framework, particularly regarding the hypergraph construction methods. For example, a more in-depth explanation of the criteria and algorithms used for defining the edges in both the Geographical Adjacency Hypergraph (GAH) and the Feature Similarity Hypergraph (FSH) would clarify how these constructions impact the model's ability to capture spatial and temporal correlations.

An elaboration on how hyperparameters were chosen during the training process, including those related to the temporal transformer (e.g., number of layers, attention heads) and the hypergraph convolution networks, would strengthen the reproducibility of the results.

**Questions:**

Could you elaborate on the self-adaptive fusion mechanism used to combine outputs from the Geographical Adjacency Hypergraph (GAH) and the Feature Similarity Hypergraph (FSH)? How are the weights learned and applied during the fusion process, and how does this help the model balance different sources of information?

Additionally, might it be beneficial to include further discussion of the technical details in the proposed framework, particularly regarding the hypergraph construction methods? For instance, could a more in-depth explanation of the criteria and algorithms used for defining the edges in both the GAH and FSH clarify how these constructions impact the model's ability to capture spatial and temporal correlations?

Lastly, can you provide an elaboration on how hyperparameters were chosen during the training process, including those related to the temporal transformer (e.g., number of layers, attention heads) and the hypergraph convolution networks? Would this strengthen the reproducibility of the results?

---

### Official Review · Reviewer_rwma · 2024-11-08

**Soundness:** 2
**Presentation:** 2
**Contribution:** 1
**Rating:** 3
**Confidence:** 5

**Summary:**

This paper presents two ways of building hypergraphs on traffic data. It employs a temporal transformer to capture temporal dependencies and utilizes hypergraph convolutional networks to inherently model spatial relationships.  The proposed method achieves good results by combining hypergraph convolution on two commonly used traffic datasets.

**Strengths:**

1.Achieved state-of-the-art results on two common datasets.

2.Open-source code is provided to reproduce their research

**Weaknesses:**

1. The method of modeling feature interactions between nodes using hypergraphs is not new, at least several works have tried this approach, but the authors did not introduce or compare it in their paper.

Dual Dynamic Spatial-Temporal Graph Convolution Network for Traffic Prediction

Tamp-S2gcnets: Coupling Time-Aware Multipersistence Knowledge Representation With Spatio-Supra Graph Convolutional Networks For Time-Series Forecasting

Metro Passenger Flow Prediction via Dynamic Hypergraph Convolution Networks

2. The baselines compared in this article are outdated. I suggest that the authors conduct more extensive experiments on a broader range of datasets to demonstrate the superiority of their proposed method over the most recent related work like STAEformer: Spatio-Temporal Adaptive Embedding Makes Vanilla Transformer SOTA for Traffic Forecasting,Heterogeneity-Informed Meta-Parameter Learning for Spatiotemporal Time Series Forecasting

**Questions:**

see the weaknesses

---

### Note · Authors · 2024-11-23

I have read and agree with the venue's withdrawal policy on behalf of myself and my co-authors.